



# Background nitrogen dioxide (NO₂) over the United States and its implications for satellite observations and trends: effects of nitrate photolysis, aircraft, and open fires

Ruijun Dang[1], Daniel J. Jacob[1], Viral Shah[1,a], Sebastian D. Eastham[2,3], Thibaud M. Fritz[2], Loretta J. Mickley[1], Tianjia Liu[4], Yi Wang[5,6], and Jun Wang[5,6]

[1] John A. Paulson School of Engineering and Applied Sciences, Harvard University, Cambridge, MA 02138, USA
[2] Laboratory for Aviation and the Environment, Department of Aeronautics and Astronautics, Massachusetts Institute of Technology, Cambridge, MA 02139, USA
[3] Joint Program on the Science and Policy of Global Change, Massachusetts Institute of Technology, Cambridge, MA 02139, USA
[4] Department of Earth and Planetary Sciences, Harvard University, Cambridge, MA 02138, USA
[5] Center for Global and Regional Environmental Research, Iowa Technology Institute, The University of Iowa, Iowa City, IA 52242, USA
[6] Department of Chemical and Biochemical Engineering, The University of Iowa, Iowa City, IA 52242, USA
[a] Now at Global Modeling and Assimilation Office, NASA Goddard Space Flight Center, Greenbelt, MD 20771, USA, and Science Systems and Applications, Inc., Lanham, MD 20706, USA

*Correspondence to:* Ruijun Dang (rjdang@seas.harvard.edu)

**Abstract.** Tropospheric nitrogen dioxide (NO₂) measured from satellites has been widely used to track anthropogenic NOₓ emissions, but its retrieval and interpretation can be complicated by the free tropospheric NO₂ background to which satellite measurements are particularly sensitive. Tropospheric NO₂ columns from the OMI satellite instrument averaged over the contiguous US (CONUS) show no trend after 2009, despite sustained decreases in anthropogenic NOₓ emissions, implying an important and rising contribution from the free tropospheric background that has not been captured in models. Here we use the GEOS-Chem chemical transport model applied to the simulation of OMI NO₂ to better understand the sources and trends of background NO₂ over CONUS. Previous model underestimate of the background is largely corrected by the consideration of aerosol nitrate photolysis, which increases the model NO₂ column by 13% on an annual basis (25% in spring), and also increases the air mass factor (AMF) to convert the tropospheric slant columns inferred from the OMI spectra into vertical NO₂ columns by 7% on an annual basis (11% in spring). The increase in the AMF decreases the retrieved NO₂ columns in the satellite observations, contributing to the improved agreement with the model. Accounting for the 2009-2017 increase in aircraft NOₓ emissions drives only a 1.4% mean increase in NO₂ column over CONUS and a 2% increase in the AMF, but the combination of decreasing surface NOₓ emissions and increasing aircraft emissions is expected to drive a 14% increase in the AMF over the next decade that will be necessary to account for in the interpretation of satellite NO₂ trends. Fire smoke identification with the NOAA Hazard Mapping System (HMS) indicates that wildfires contribute 1-



8% of OMI NO$_2$ columns over the western US in June-September and that this contribution has been increasing since 2009, contributing to the flattening of OMI NO$_2$ trends. Future analyses of NO$_2$ trends from satellite data to infer trends in surface NO$_x$ emissions must critically consider the effects of a rising free tropospheric background due to increasing emissions from aircraft, fires, and possibly lightning.

## 1 Introduction

Nitrogen oxides (NO$_x$ ≡ NO + NO$_2$) emitted by combustion, lightning, and soils affect oxidant chemistry, air quality, climate, and ecosystems. NO$_x$ cycles chemically to produce tropospheric ozone and is eventually oxidized to nitric acid (HNO$_3$), which partitions into the particulate phase and is removed by deposition. Satellite observations of tropospheric NO$_2$ columns have been used extensively to infer anthropogenic NO$_x$ emissions and their changes (Martin et al., 2003; Boersma et al., 2008; Russell et al., 2012; Duncan et al., 2016; Miyazaki et al., 2017; Matthew J. Cooper et al., 2020; Wang et al.,

2020). However, recent studies for North America suggest that the free tropospheric background NO$_2$ (above ~2 km altitude) could make a large and increasing contribution to the tropospheric NO$_2$ columns measured from space (Silvern et al., 2019; Zhu et al., 2019; Qu et al., 2021; Y. Wang et al., 2021; Jiang et al., 2022). Here we investigate the sources of this NO$_2$ background over the US and the implications for the retrieval and interpretation of satellite observations.

Tropospheric NO$_2$ has been measured from space by solar backscatter since 1995 with the GOME instrument (Martin et al., 2002). Current instruments include OMI (2004-) (Krotkov et al., 2016), OMPS (2012-) (Yang et al., 2014), TROPOMI (2017-) (van Geffen et al., 2020; Li et al., 2022), and the geostationary GEMS over East Asia (2020-) (Kim et al., 2020). Tropospheric NO$_2$ is retrieved in three steps: (1) fit the measured backscattered solar spectrum in and around the NO$_2$ absorption band (400-470 nm) to infer a slant column density (SCD) along the optical path; (2) subtract the stratospheric

contribution from the SCD to obtain the tropospheric SCD; (3) convert the tropospheric SCD to a tropospheric vertical column density (VCD) by dividing by an air mass factor (AMF). The AMF accounts for the photon path between the Sun and the satellite instrument in relation to the vertical distribution of NO$_2$. It is calculated by convolving altitude-dependent detection sensitivities (scattering weights) computed from a radiative transfer model with a local NO$_2$ normalized vertical profile (shape factor) provided by an independent chemical transport model (CTM) (Palmer et al., 2001). The detection

sensitivity typically increases by a factor of 5 from the surface to the upper troposphere because of atmospheric scattering (Martin et al., 2002; Boersma et al., 2016; M. J. Cooper et al., 2020), so that the tropospheric column measurement can be most sensitive to free tropospheric NO$_2$ even under fairly polluted conditions (Travis et al., 2016). The CTM must then properly represent this free tropospheric NO$_2$ background. Although cloud-slicing methods have been used to isolate the free tropospheric contribution in satellite NO$_2$ retrievals, these products have large errors and show large inconsistencies (Choi et

al., 2014; Belmonte Rivas et al., 2015; Marais et al., 2018; Marais et al., 2021).



The importance of characterizing the free tropospheric $NO_2$ background was brought to the fore by the use of OMI $NO_2$ data to infer $NO_x$ emission trends in the contiguous United States (CONUS) (Jiang et al., 2018; Silvern et al., 2019; Qu et al., 2021; Y. Wang et al., 2021; He et al., 2022; Jiang et al., 2022). The OMI $NO_2$ data over CONUS show a steady decrease

from 2005 to 2009 (Russell et al., 2012; Duncan et al., 2013; Duncan et al., 2016; Krotkov et al., 2016), consistent with the decreases of $NO_x$ emissions reported in the EPA National Emission Inventory (NEI), but Jiang et al. (2018) found that the trend flattened after 2009 despite sustained decreases in $NO_x$ emissions according to the NEI and supported by $NO_2$ surface data (Silvern et al., 2019). This flattening of the trend was attributed by Silvern et al. (2019) to an increasing relative contribution to the OMI $NO_2$ column from the free tropospheric background as the anthropogenic $NO_x$ emissions decrease,

and further evidence for this emerged from the COVID-19 economic shutdown in 2020 when the response of OMI observations to the decrease of $NO_x$ emissions in CONUS was much less than the response of surface $NO_2$ observations (Qu et al., 2021). Flattening of the OMI $NO_2$ trend despite continuing decrease in $NO_x$ emission would actually require an absolute increase in the background, the source of which is not yet clear (Jiang et al., 2022).

Sources of background free tropospheric $NO_x$ include lightning, aircraft, buoyantly lofted fire plumes, convective injection from the boundary layer, and downwelling of stratospheric air, all further complicated by chemical cycling with nitrates and by long-range transport (Lamarque et al., 1996; Schumann, 1997; Jaegle et al., 1998; Val Martín et al., 2006; Hudman et al., 2007; Zhang et al., 2012; Zhu et al., 2019). Lightning accounts for ~80% of $NO_x$ in the upper troposphere (UT, above 8 km) in warm seasons (Hudman et al., 2007; Zhu et al., 2019), and aircraft is the dominant UT source in northern winter

(Schumann, 1997). Fire emissions can be injected by buoyancy above the boundary layer (Val Martin et al., 2018) and lead to enhanced $NO_x$ in the free troposphere (Val Martín et al., 2006; Reidmiller et al., 2010). Y. Wang et al. (2021) suggested that lightning and soil $NO_x$ emissions over CONUS had a downward trend during 2005-2009 followed by an upward trend during 2009-2019, partly explaining the post-2009 flattening of OMI $NO_2$.

Aircraft and wildfires could also contribute to the background $NO_2$ trend but this has not been studied so far. Aircraft emissions increased at a rate of 3.3% per year over the past decade (Lee et al., 2021). Aircraft emission inventories in the CTMs used for satellite retrievals tend to be outdated and do not account for this rapid growth (Boersma et al., 2018; Lamsal et al., 2021). Large $NO_2$ enhancements have been detected by satellites at fire locations (Mebust et al., 2011; Griffin et al., 2021; Jin et al., 2021), but little is known about the more general contribution of fires to the tropospheric $NO_2$ background. A

recent study reported that 19-56% of peroxyacetyl nitrate (PAN) detected in the free troposphere over the western US in summer 2018 by the Cross-Track Infrared Sounder (CrIS) was associated with fire smoke (Juncosa Calahorrano et al., 2021). Wildfires have become increasingly frequent and intense in the US over the past two decades (Westerling, 2016; Jaffe et al., 2020) and the fire season has lengthened (Cattau et al., 2020), both of which would contribute to an increase in background $NO_2$.




Here we aim to better understand the sources and trends of the free tropospheric background NO$_2$ (and more generally background NO$_x$) in CONUS by (1) investigating the role of aerosol nitrate photolysis as a source of NO$_x$ (Shah et al., 2022), (2) quantifying the effect of increasing aircraft emissions with the Aircraft Emissions Inventory Code (AEIC; Simone et al., 2013), and (3) quantifying wildfire influence with the NOAA Hazard Mapping System (HMS) smoke product (Rolph et al., 2009) in relation to the OMI NO$_2$ observations. We use observations from OMI to evaluate the ability of the GEOS-Chem chemical transport model (CTM) to simulate tropospheric NO$_2$ columns over CONUS in different seasons, and examine the implications for improving the shape factors and their trends in satellite retrievals.

## 2 GEOS-Chem model

We use the GEOS-Chem global CTM version 13.1.2 (https://doi.org/10.5281/zenodo.5014891) driven by MERRA-2 meteorology (Gelaro et al., 2017) in a simulation of oxidant-aerosol chemistry over CONUS in 2009 and 2017. The simulation is at the native MERRA-2 0.5° × 0.625° resolution over North America (10–70° N, 140–40° W), nested in a 4° × 5° global simulation with boundary conditions updated every 3 h. GEOS-Chem 13.1.2 includes a detailed representation of chemistry with recent updates for NO$_x$ uptake by aerosols and clouds (Holmes et al., 2019), isoprene chemistry (Bates and Jacob, 2019), and halogen chemistry (X. Wang et al., 2021). Dry deposition follows a standard resistance-in-series scheme (Wesely, 1989), with HNO$_3$ updates from Jaeglé et al. (2018). Wet deposition includes in-cloud rainout, below-cloud washout, and scavenging in convective updrafts (Liu et al., 2001). For HNO$_3$, we use a faster washout rate described in Luo et al. (2019), following previous applications for China (Zhai et al., 2021) and remote oceans (Travis et al., 2020).

Table 1 gives the CONUS NO$_x$ emissions in the model for 2009 and 2017. Global anthropogenic emissions for individual years are from the Community Emissions Data System (CEDS) (McDuffie et al., 2020). This is superseded for the US by the EPA NEI2016 inventory for 2016 (https://www.epa.gov/air-emissions-modeling/2016v1-platform), scaled to 2009 and 2017 using national emission totals (EPA, 2021). Soil NO$_x$ emissions are from Hudman et al. (2012) and updated by Y. Wang et al. (2021) to allow for increased soil NO$_x$ emissions under high temperature conditions (30-40° C) and using soil temperatures from MERRA-2 instead of air temperatures. We reduced the summertime soil NO$_x$ emissions in the midwestern US (38-50° N, 105-95° W) by 50%, as suggested by a previous comparison with OMI NO$_2$ observations (Vinken et al., 2014). Lightning NO$_x$ emissions are computed following Murray et al. (2012), with lightning flash density spatially constrained by Lightning Imaging Sensor/Optical Transient Detector (LIS/OTD) climatology, interannual variation driven by convective cloud height from MERRA-2, and vertical distribution following Ott et al. (2010). Monthly open fire emissions are taken from the Global Fire Emissions Database version 4 with small fires (GFED4s) (van der Werf et al., 2017), with all releases at the surface. Interannual variations in GFED4s are driven by changes in fuel load, which is a function of meteorology and vegetation type, and burned area and active fire data derived from MODIS satellite observations. Not included in Table 1 is the source from cross-tropopause transport of reactive nitrogen oxides produced in





the stratosphere by oxidation of $N_2O$. This source is included in the model but is negligibly small (0.2 Tg N $a^{-1}$ globally; Bey et al., 2001).


**Table 1.** Contiguous US (CONUS) $NO_x$ emissions in 2009 and 2017 [Tg N $a^{-1}$] [a].

|  | 2009 | 2017 |
|---|---|---|
| Total | 4.9 | 3.6 |
| Fuel combustion [b] | 3.8 | 2.3 |
| Lightning | 0.43 | 0.48 |
| Soil | 0.55 | 0.60 |
| Aircraft | 0.13 | 0.16 |
| Open fires | 0.03 | 0.10 |

[a] as used in the GEOS-Chem simulations described in the text.

[b] excluding aircraft.

For aircraft emissions, we use an updated inventory for 2019 produced by the Aircraft Emissions Inventory Code (AEIC) developed at MIT (Simone et al., 2013). It includes daily emissions during both landing and take-off (LTO) and at cruise with horizontal resolution of 0.5° × 0.625° and vertical resolution of 36 layers up to 100 hPa. 81% of these emissions are released above 5 km altitude, and 73% above 8 km. AEIC2019 is scaled to individual years and for different continental regions by using the kerosene consumption data provided by the International Energy Agency (IEA,

https://www.iea.org/fuels-and-technologies/oil). The $NO_x$ emission factor increased by 7% from 2005 to 2019 due to increase in combustor temperatures (Dedoussi, 2021; Lee et al., 2021). As shown in Figure 1, aviation $NO_x$ emissions increased globally by 51% from 2005 to 2019, largely driven by the rapid growth in developing countries such as China, and by 17% for CONUS. The dip in 2007-2009 is due to the economic recession (Bows-Larkin et al., 2016). AEIC2019 may underestimate aircraft emissions by 10-40% because it does not consider military aircraft, estimated to account for 10-13%

of total emissions (Wilkerson et al., 2010) and because it assumes great circle routes which could underestimate emission by up to 28% (Zhang et al., 2022). The effect of aircraft emissions on upper tropospheric $NO_x$ in GEOS-Chem may further be underestimated by 10% due to nonlinear aircraft plume chemistry increasing the $NO_x$ lifetime (Fritz et al., 2020).



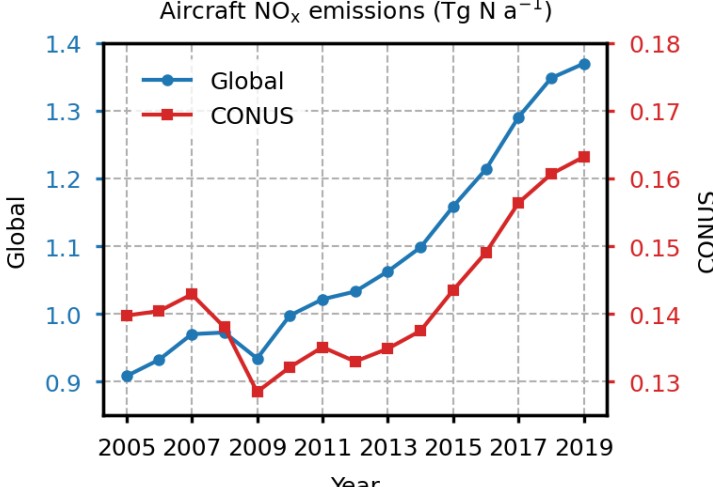

**Figure 1:** Global and CONUS aircraft NO$_x$ emissions in the AEIC2019 inventory from 2005 to 2019 (Lee et al., 2021).

Table 1 shows how the trends in NO$_x$ emissions over CONUS from 2009 to 2017 would drive an increasing contribution from the background to the satellite observations of NO$_2$ columns. The anthropogenic fuel combustion source (excluding aircraft) accounted for 78% of total NO$_x$ emissions in 2009 but only 64% in 2017. It decreased by 39% from 2009 to 2017, but total NO$_x$ emissions decreased by only 23% as background NO$_x$ emissions increased by 18% with all sources contributing to this increase (lightning, soils, aircraft, open fires). There is interannual variability in these background sources that we will discuss below for fires. Nevertheless, factoring in the greater sensitivity of satellite observations to the free troposphere, we see how a combination of reduced anthropogenic emissions and increasing background could lead to a flattening of the tropospheric NO$_2$ columns observed from space.

We also add in our simulation the photolysis of particulate nitrate (pNO$_3^-$) previously introduced in GEOS-Chem by Kasibhatla et al. (2018) and Shah et al. (2022) to correct model underestimates of tropospheric NO$_x$. Gas-phase HNO$_3$ can convert back to NO$_2$ through photolysis or reaction with OH, but this process is slow with a lifetime of 15-30 days in the troposphere (Dulitz et al., 2018). Previous studies have reported a 10-300 times faster photolysis rate of nitrate in the aerosol phase (Ye et al., 2016; Reed et al., 2017; Ye et al., 2017), particularly in the presence of chloride (Wingen et al., 2008; Richards et al., 2011; Zhang et al., 2020), recycling NO$_x$ through two branches producing NO$_2$ and nitrous acid (HONO). This process has been found to be an important missing source of NO$_x$ over the remote oceans (Ye et al., 2016; Reed et al., 2017; Kasibhatla et al., 2018). Shah et al. (2022) found that incorporating aerosol nitrate photolysis in the GEOS-Chem model largely corrects the model's underestimation of NO$_x$ over the oceans during the ATom aircraft campaign. Here we follow the approach of Shah et al. (2022), who use equation (1) to calculate the enhancement factor (EF) of pNO$_3^-$ photolysis



relative to the HNO₃ photolysis frequency, as a function of the molar concentrations of sea salt aerosol [SSA] and nitrate [pNO₃⁻]:

$$EF = 100 * \max\left(\frac{[SSA]}{[SSA]+[pNO_3^-]}, 0.1\right), \tag{1}$$

The resulting EF values range from 10 in continental air with low SSA to 100 in marine air, and a 2:1 branching ratio for the
HONO:NO₂ branches is assumed.

Figure 2 shows the effect of pNO₃⁻ photolysis as described above on the simulated NO₂ vertical profiles over CONUS. There is no significant effect in the boundary layer (BL) below 2 km altitude because other sources dominate but concentrations in the free troposphere increase by 25% on an annual basis. The largest increase is in spring due to a combination of high pNO₃⁻
abundance and strong radiation. The implications for simulation of OMI observations are discussed in Section 3.

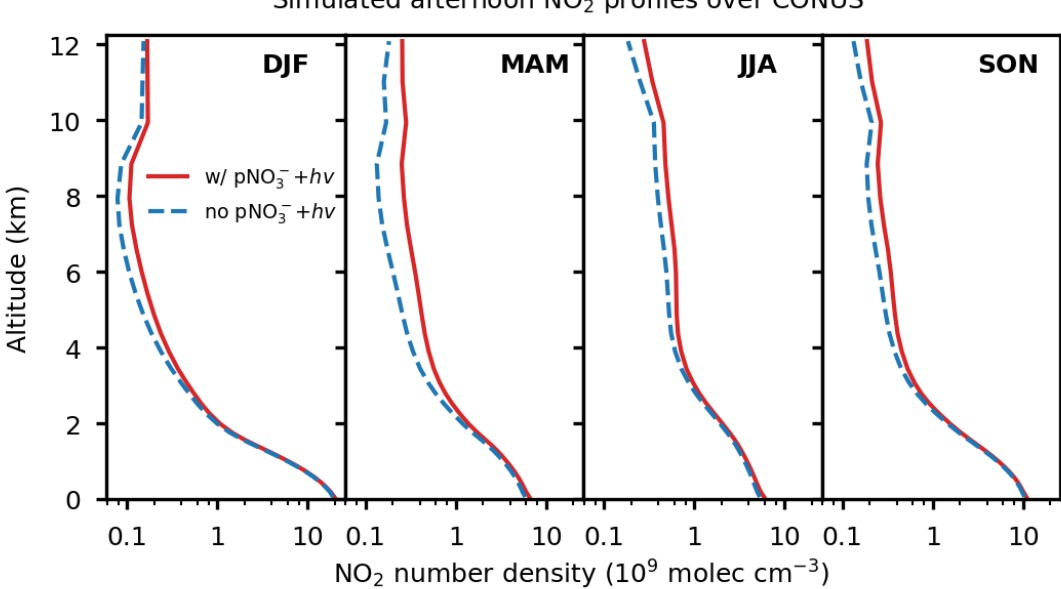

**Figure 2**: Mean vertical profiles of NO₂ number density simulated by GEOS-Chem over the contiguous US (CONUS) in 2017 at the OMI overpass time of 13-14 local time (LT) and in different seasons. Solid/dashed lines indicate the NO₂ profiles
with/without particulate nitrate photolysis (pNO₃⁻ + $hv$). Note log scale for abscissa.



## 3 Effect of nitrate photolysis

Figure 3 compares OMI observed and GEOS-Chem simulated tropospheric NO$_2$ columns over CONUS in 2017. The OMI instrument onboard the Aura satellite provides daily global coverage at 13:30 local time (LT) with 13x24 km$^2$ nadir pixel

resolution. Here, we use NO$_2$ retrievals from version 4.0 of the NASA OMI NO$_2$ level 2 product (OMNO2; https://disc.gsfc.nasa.gov/datasets/OMNO2_003/summary) (Lamsal et al., 2021), filtered by removing pixels with cloud fraction >0.2, surface reflectivity >0.3, solar zenith angle >75°, view zenith angle >65°, as well as pixels affected by the so-called row anomaly (Dobber et al., 2008). The OMNO2 product uses vertical shape factors from the Global Modeling Initiative (GMI) CTM in its AMF calculation. To conduct a consistent comparison with GEOS-Chem, we recompute the

AMFs and hence the OMI retrievals (now referred to as OMI-GC) using local GEOS-Chem NO$_2$ shape factors sampled at 13-14 LT (Boersma et al., 2016; M. J. Cooper et al., 2020).

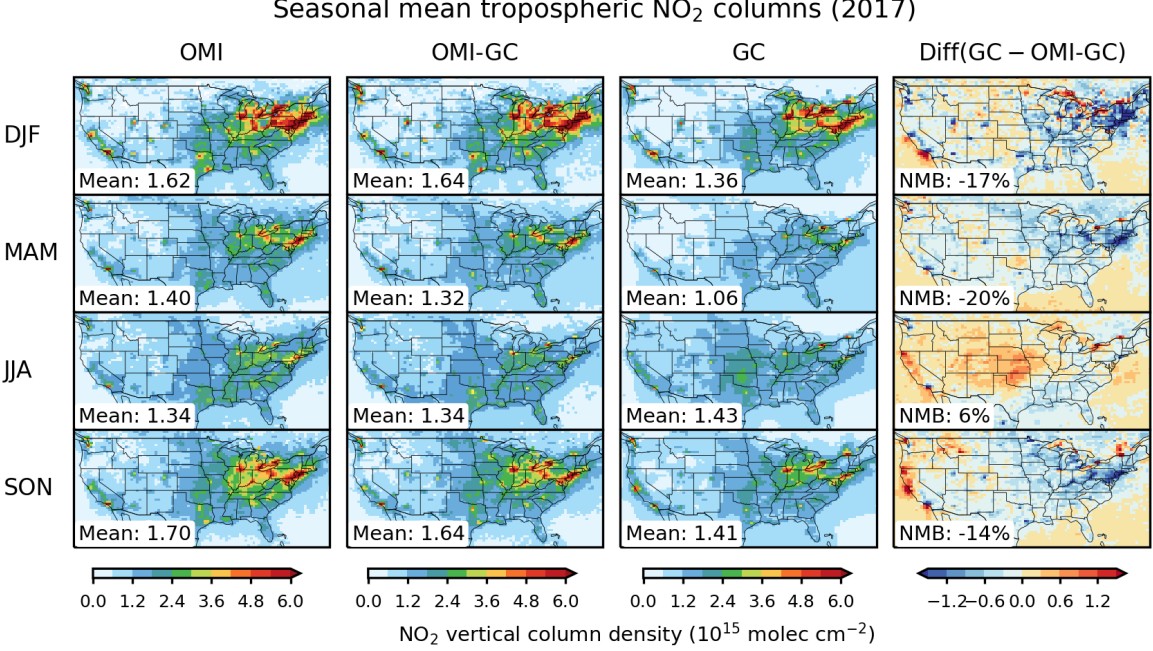

**Figure 3:** Seasonal mean tropospheric NO$_2$ columns over CONUS in 2017. From left to right are the OMNO2 OMI retrievals

with AMFs computed using the GMI NO$_2$ vertical profiles (OMI), the OMI retrievals with AMFs computed using the GEOS-Chem NO$_2$ vertical profiles (OMI-GC), the GEOS-Chem simulation (GC), and the difference between GC and OMI-GC. Mean values over CONUS and normalized mean bias (NMB) between GC and OMI-GC are given inset.

GEOS-Chem generally captures the observed spatial distribution and seasonal variation of tropospheric NO$_2$ columns in

CONUS. The normalized mean bias (NMB) is -17% in winter, -20% in spring, +6% in summer, and -14% in fall. In





comparison, the NMB without nitrate photolysis is -24% in winter, -43% in spring, -14% in summer, and -26% in fall. Adding nitrate photolysis greatly improves the agreement, and its effect on NO₂ columns and AMFs is shown in Figure 4. It increases simulated tropospheric NO₂ columns by 13% over CONUS, with the largest increase in spring (25%). In figure 4b, we calculate the monthly AMFs by using scattering weights from OMNO2 v4 and GEOS-Chem NO₂ shape factors with and

without nitrate photolysis to determine the effect of nitrate photolysis on NO₂ retrievals. Annual mean AMF increases by 7% and spring AMF increases by 11% with nitrate photolysis because the shape factor is shifted to higher altitudes (Figure 2). As a result, the NO₂ column retrieved using GEOS-Chem NO₂ shape factor (OMI-GC) with nitrate photolysis is 7% lower than that retrieved using the shape factor without nitrate photolysis (Figure 4a), and this further improves agreement with the model simulation. The annual mean AMF using the GEOS-Chem shape factor with nitrate photolysis is now 1.21, closer to

the AMF calculated using the GMI shape factor (1.19). GMI does not include nitrate photolysis but has slower NOₓ loss from N₂O₅ chemistry than GEOS-Chem (Shah et al., 2022).



**Figure 4:** Sensitivity to particulate nitrate photolysis of tropospheric NO₂ column densities simulated by GEOS-Chem and

retrieved from OMI satellite observations over CONUS. The top panel shows 2017 monthly mean GEOS-Chem NO₂ columns with and without nitrate photolysis, and the OMI retrieval using GEOS-Chem shape factors (OMI-GC) with and



without nitrate photolysis. The bottom panel shows the monthly mean air mass factor (AMF) for OMI tropospheric $NO_2$ retrievals with and without nitrate photolysis. Numbers inset give annual mean percentage differences for the different quantities with versus without nitrate photolysis.


Overall, our GEOS-Chem simulation including $pNO_3^-$ photolysis is consistent with the OMI observations within their 35% estimated uncertainty (Lamsal et al., 2021), reflecting the combined and opposite effects of $pNO_3^-$ photolysis on the simulated $NO_2$ columns and on the satellite retrieval. Figure 3 still shows some regional discrepancies between GEOS-Chem and OMI. Low biases are found in urban areas that may reflect biases in model chemistry at 50 km resolution but also bias in the satellite $NO_2$ retrievals (Laughner and Cohen, 2019; Laughner et al., 2019). There is a high bias in the central US during the summer, even before the inclusion of nitrate photolysis, partly due to the use of the updated soil $NO_x$ scheme of Y. Wang et al. (2021) in GEOS-Chem that increases emissions at high temperatures (Y. Wang et al., 2021). On the west coast, high biases are found in fire-influenced areas but that may be due to OMI retrievals being too low. Most current operational $NO_2$ retrieval algorithms including OMNO2 v4 treat aerosols implicitly (Liu et al., 2020; Vasilkov et al., 2021), resulting in low retrieved $NO_2$ in fire plumes (Griffin et al., 2021). Travis et al. (2016) previously found a 30% GEOS-Chem overestimate of the NASA OMI $NO_2$ v2.1 retrieval over the Southeast US in summer, which they attributed to the NEI2011 emissions being too high, but that model bias is largely corrected in our simulation due both to downward revision of $NO_x$ emissions in NEI2016 and to 10-40% higher OMI $NO_2$ retrievals in version 4 of OMNO2 relative to previous versions (Lamsal et al., 2021).

## 4 Effect of aircraft emissions

We quantify the effect of the 2009-2017 increase in aircraft emissions (Fig. 1) on $NO_2$ columns and AMFs over CONUS by conducting sensitivity simulations with the AEIC emission trends, either taken at face value or increased by 50% as discussed in Section 2, and all other conditions, including anthropogenic surface emissions, fixed for 2017. The increase in aircraft emissions since 2009 leads to a 1.4-2.0% increase in the simulated annual mean tropospheric $NO_2$ columns over CONUS, with 75% of the increase concentrated above 6 km. The AMF increases by 1.7%-2.3% and there is a corresponding decrease in the retrieved $NO_2$ columns, which is not considered in the OMI retrievals. Previous studies found that GEOS-Chem could not fully reproduce the flattening of the $NO_2$ trend over CONUS seen by OMI (Silvern et al., 2019; Qu et al., 2021). Increasing aircraft emissions is a small but significant effect to correct this discrepancy, reducing by 13-16% the difference between modeled and retrieved trends over the 2009-2017 period.


The increase in aircraft emissions, combined with the decrease of anthropogenic surface emissions, leads to a 10% increase in the annual mean AMF over CONUS from 2009 to 2017 (Figure 5) as the vertical distribution of $NO_2$ shifts from the boundary layer to the upper troposphere. This effect is expected to increase in the future, as shown in Figure 5 with a




projection to 2032. In this projection, anthropogenic emissions over CONUS are assumed to continue to decrease at the

current rate of 5.9% a$^{-1}$. Global aircraft emissions are 20% lower in 2022 than in 2019 due to the pandemic. We assume they return to 2019 levels in 2023, and increase at a rate of 3.6% a$^{-1}$ in the following decade (Airbus, 2022; Boeing, 2022). We calculate the resulting AMFs by applying scattering weights from OMNO2 v4 to the extrapolated NO$_2$ vertical profiles and find an AMF increase of 20% over the 2017-2032 period, including 80% from decrease in surface emissions and 20% from increase in aircraft emissions. Increasing aircraft emissions will likely play an increasing role in increasing the AMF in the

future.

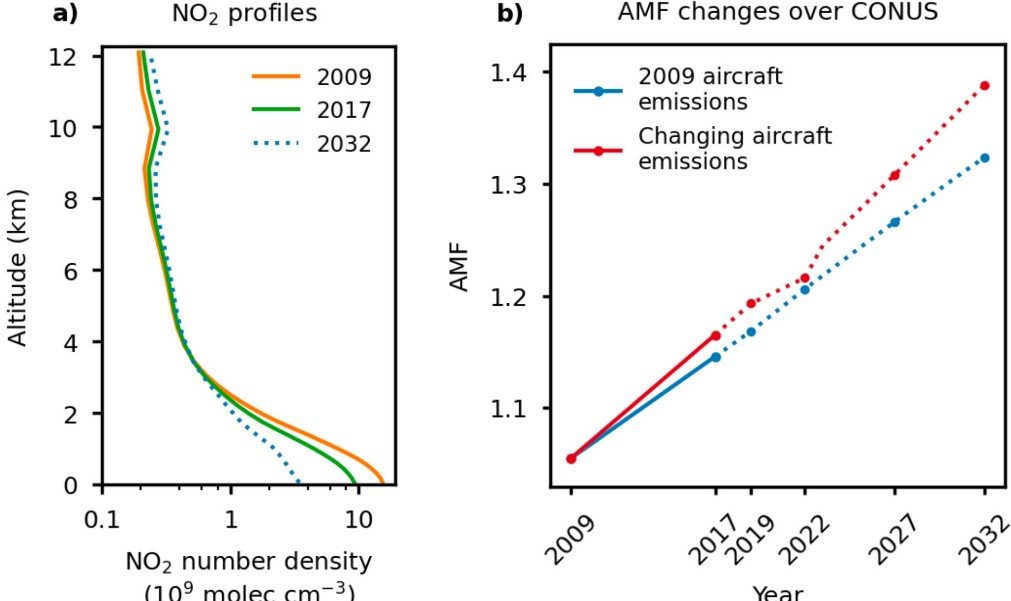

**Figure 5:** Effect of changing emissions on tropospheric NO$_2$ shape factors and air mass factors (AMFs) for tropospheric NO$_2$ retrievals over CONUS. a) NO$_2$ afternoon profiles over CONUS in 2009, 2017, and 2032 (projected), accounting for

changes in surface and aircraft emissions. b) AMFs for tropospheric NO$_2$ retrievals over CONUS from 2009 to 2032, with the red line calculated using NO$_2$ profiles with reduced anthropogenic surface emissions and increased aircraft emissions, and the blue line with reduced anthropogenic surface emissions only and fixed aircraft emissions for 2009. Solid lines are from GEOS-Chem simulations for 2009 and 2017, while dashed lines scale the 2009-2017 NO$_2$ simulation differences to the projected changes in emissions beyond 2017.




## 5 Effect of open fire emissions

Open fire emissions make a very small contribution to the mean CONUS NOx budget in GEOS-Chem (Table 1), but they could be of seasonal importance in the western US in June-September (Jaffe et al., 2020). Simulation of fire NOx in models such as GEOS-Chem is affected by uncertainties not only in emissions (Carter et al., 2020) but also in plume lofting and

chemistry (Zhu et al., 2018; Palm et al., 2021; Peng et al., 2021). Interpreting fire $NO_2$ observations by satellites is complicated by scattering and absorption from the smoke aerosols (Griffin et al., 2021) as well as plume lofting (Jin et al., 2021) and any differences in vertical distribution between $NO_2$ and the smoke aerosols.

Here we use the Hazard Mapping System (HMS) product (Rolph et al., 2009) from the National Oceanic and Atmospheric

Administration (NOAA) to identify fire-impacted OMI retrievals and estimate from there the contribution of fire to $NO_2$ columns as observed by OMI. HMS provides daily daytime locations and extent of smoke plumes as determined by human analysts using satellite imagery to delineate smoke-affected areas. The dataset has been widely used to identify such areas (Brey and Fischer, 2016; Fischer et al., 2018; Juncosa Calahorrano et al., 2021). We use the fire-season (June-September) HMS data from 2009 to 2020 (https://www.ospo.noaa.gov/Products/land/hms.html#data) and grid it to a horizontal

resolution of 0.25°x0.25°. For each grid cell, we define days associated with smoke plumes in the HMS data as fire-affected days and the remaining days as fire-free days. Fire-free days may still have some influence of aged fire emissions such as through PAN decomposition, so the distinction offered by the HMS data is for relatively fresh fire emissions over a few days. We thus infer the contribution of fresh fire emissions to the $NO_2$ columns from the difference between the seasonal mean $NO_2$ columns and the average $NO_2$ columns for fire-free days. Previous studies have found that the implicit correction

for aerosols in current $NO_2$ retrieval algorithms as effective clouds could introduce low biases of up to 50% in the retrieved $NO_2$ column in areas of high aerosol loading, such as fire plumes (Lorente et al., 2017; Liu et al., 2020). Therefore, we also examine the effect of increasing tropospheric $NO_2$ columns on fire-affected days by 50%.

Figure 6 shows the contribution of fresh fire emissions to the OMI observations over the western US (130°-114° W, 30°-50°

N) and the rest of CONUS in June-September 2009-2020. The red envelope shows the effect of increasing tropospheric $NO_2$ columns on fire-affected days by 50% to account for aerosol-induced bias, and this defines the upper range of fire influence. We find that fires contribute 1-8% (upper range 1-15%) to the OMI $NO_2$ columns over the western US in individual years, with a maximum contribution in 2020 which was a particularly high fire year. The fire contribution in the rest of CONUS is negligible. In the absence of fires, we see from Figure 6 that OMI $NO_2$ over the western US would have significantly

decreased over the 2009-2020 period. However, increasing fire activity over the past decade has flattened the OMI trend. We see no such effect over the rest of CONUS, where open fires have negligible influence, and increasing lightning could be responsible for the flattening of OMI $NO_2$ there (Y. Wang et al., 2021).



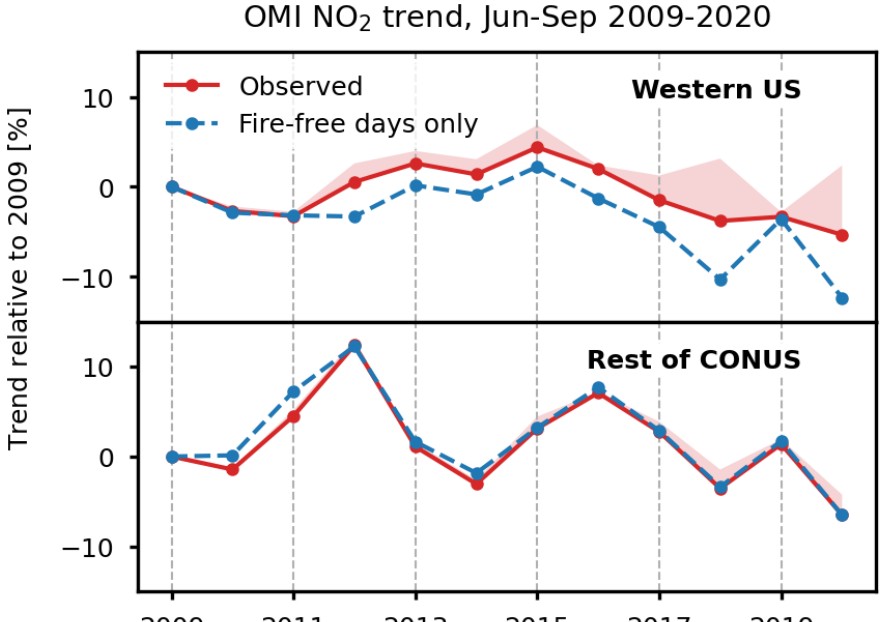

**Figure 6:** Trends of OMI tropospheric NO₂ columns over the western US and the rest of CONUS for June-September 2009-2020, examining the effects of fires. Trends are shown relative to 2009. The overall trend from the OMNO2 retrieval is shown in red, and the trend for the subset of fire-free days is shown as dashed. The red envelope shows the effect of a 50% bias correction of the OMI NO₂ retrieval on fire-affected days. The western US domain is defined as (130°-114° W, 30°-50° N).

## 6 Conclusions

OMI satellite observations of tropospheric NO₂ columns over the contiguous US (CONUS) show a flattening trend after 2009 despite continuous decrease in surface anthropogenic NO$_x$ emission. This suggests a large and rising contribution from background NO₂ in the free troposphere (above 2 km) to the tropospheric NO₂ columns observed from space, but previous simulations with the GEOS-Chem chemical transport model have been unable to account for this background and its increase. Here, we used GEOS-Chem including a new mechanism for particulate nitrate photolysis and the AEIC2019 inventory for aircraft emission trends, together with the NOAA Hazard Mapping System (HMS) smoke product, to better understand the magnitude and trends of free tropospheric background NO₂ and the implications for retrieval and interpretation of satellite data.

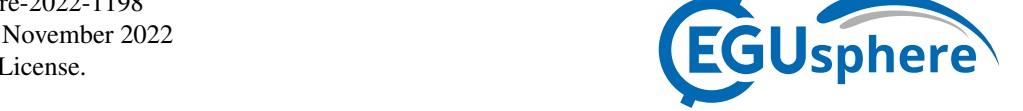

Inclusion of particulate nitrate photolysis as a secondary source of $NO_x$ in GEOS-Chem, following recent field and laboratory evidence, increases the annual mean tropospheric $NO_2$ columns over CONUS by 13% with the maximum effect in spring (25%). It also increases the air mass factor (AMF) for the satellite tropospheric $NO_2$ retrieval by 7% on an annual basis (11% in spring) because of the increased contribution of the free troposphere to the tropospheric $NO_2$ column. The combination of these two effects provides an improved fit of GEOS-Chem to the OMI satellite $NO_2$ data, particularly in the spring when the model underestimate was worst.

Aircraft $NO_x$ emissions increased by 38% globally from 2009 to 2017 and by 20% over CONUS but this has generally not been taken into account in the models used to derive vertical shape factors for satellite $NO_2$ retrievals. The associated increase in the annual mean AMF is only 2% because aircraft emissions remain relatively low compared to lightning. Nevertheless, we find that consideration of the 2009-2017 aircraft emission trend corrects 13-16% of the discrepancy between GEOS-Chem and OMI $NO_2$ trends over that period, again due both to the increase in GEOS-Chem $NO_2$ and to the AMF-induced decrease in the OMI $NO_2$ retrieval. Sustained increases in aircraft emissions in the future, combined with sustained decreases in surface anthropogenic emissions, are expected to increase the AMF by 14% over the next decade with major implications for interpretation of $NO_2$ trends from satellite data.

The contribution of open fires to the free tropospheric $NO_2$ background is difficult to diagnose in GEOS-Chem due to uncertainties in plume lofting and chemistry. Instead, we used the NOAA Hazard Mapping System (HMS) to separate fire-free from fire-affected days in the OMI $NO_2$ data during the June-September fire season in the western US. We find that fires contribute only 1-8% of the seasonal OMI $NO_2$ in that region for individual years, but with an increasing trend over 2009-2017 that could offset the decrease in anthropogenic emissions and further explain the flattening of the OMI $NO_2$ tropospheric column trend during that period.

Our work demonstrates the importance of properly accounting for the free tropospheric background in interpreting $NO_2$ observations from space. Better understanding is needed of the role of aerosol nitrate photolysis, which remains highly uncertain. Increasing contributions to the tropospheric $NO_2$ column from free tropospheric sources including aircraft and fires, and possibly lightning (Y. Wang et al., 2021), combined with decrease in surface anthropogenic emissions, will be critical to consider in future analyses of $NO_2$ trends from satellite data.

**Data availability**

The OMNO2 product are available at https://disc.gsfc.nasa.gov/datasets/OMNO2_003/summary. The HMS data can be downloaded from https://www.ospo.noaa.gov/Products/land/hms.html#data.The AEIC2019 inventory is now in the default GEOS-Chem version13.4. All other model results are available on request from the corresponding author.



**Author contributions**

DJJ and RD designed the study. RD conducted model simulations and analyzed satellite and model data. VS contributed nitrate photolysis in GEOS-Chem and supported data analysis. SDE and FT provided aircraft emission inventory and
supporting guidance. JW and YW contributed soil $NO_x$ emission scheme in GEOS-Chem. LJM and TL helped with interpretation and discussion related to HMS product. RD and DJJ wrote the manuscript and all authors contributed to the revision of the paper.

**Competing interests**

The authors declare that they have no conflict of interest.

**Acknowledgements**

We acknowledge the Global Modeling and Assimilation Office (GMAO) at NASA Goddard Space Flight Center for providing the MERRA-2 data.

**Financial support**

This work was supported by the NASA Aura Science Team and by the US EPA Science to Achieve Results (STAR)
Program.

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
