# Peer review of "Background nitrogen dioxide (NO2) over the United States and its implications for satellite observations and trends: effects of nitrate photolysis, aircraft, and open fires"

_EGUsphere, 2022_

## Author Response (AR1)

**Response to comments by Referee #1**

The study by Dang et al follows up on the ongoing conversation on the importance of free tropospheric NO₂ for satellite retrievals and their interpretation. The paper puts forward interesting hypotheses and sensitivity tests for tropospheric NO₂ that can be seen as guidelines for future research. The prediction of how satellite UV/Vis air mass factors may change in response to increasing aircraft NOₓ emissions is exciting, and provides a clear message to the retrieval community to stay on their toes in accounting for possible changes in atmospheric composition.

The paper is very well written and clear in reporting its model-based findings and providing recommendations. The one difficulty I have with judging the relevance of the findings is that many of the statements rely on the confidence the authors have in the accuracy of their baseline simulations of FT NO₂. The paper would be much stronger if the authors would evaluate their simulated FT NO₂ profiles with aircraft NO₂ profiles that have been measured over the contiguous US over the last two decades via campaigns, or with simulations from other chemistry transport models (e.g. GMI). Such a comparison could help to substantiate the claim that including an NO₂ source from nitrate photolysis in the GEOS-Chem simulation can explain 'a large and increasing contribution from background NO₂ in the free troposphere'.

Another question is why the effect of enhanced NO₂ from nitrate photolysis is present so ubiquitously throughout the free troposphere. I would think that the nitrate aerosol vertical distribution has -on average- lowest mass density in the upper troposphere and highest mass density in the lower troposphere (above the boundary layer), and therefore constitutes a stronger source of NO₂ in the mid-troposphere than in the upper troposphere. But the NO₂ increase from nitrate photolysis shows up throughout the entire free troposphere. Could this be related to stronger radiation levels in the UT? Do the authors also see a spatial gradient in the NO₂ addition with stronger NO₂ additions in the warm seasons, and close to the coasts?

We thank the reviewer for their thoughtful and supportive comments. Our response to the main concerns and specific comments is as follows:

**Main concerns:**

*Evaluation of FT NO₂ profiles:* A recent study (Shah et al., 2023) has comprehensively evaluated the GEOS-Chem simulations of tropospheric NO₂ profiles using aircraft campaigns (SEAC⁴RS, DC3, and ATom) and three other atmospheric chemistry models (GMI, TM5, and CAMS), and has shown that GEOS-Chem with nitrate photolysis successfully reproduces the shape of tropospheric NO₂ for SEAC⁴RS and DC3 over the US. We have now included this in the revised manuscript.

*Cause of ubiquitous FT NO₂ addition from nitrate photolysis:* The x-axis of Figure 2 is in log scale, so it appears that the NO₂ addition from nitrate photolysis is greater in the upper troposphere (UT) than in the lower troposphere (LT), while in fact, nitrate photolysis leads to a greater increase of NO₂ in the LT than in the UT (see Figure R1 below). The vertical shape of NO₂ addition is determined by a combination of the pNO₃⁻ shape and the J(pNO₃⁻) shape. We have clarified this in the revised manuscript. *Seasonal and spatial variation in NO₂ addition:* NO₂ addition is not necessarily stronger in warm seasons or near the coast, as it is determined by a combination of pNO₃⁻ concentration, radiation, and enhancement factor. For example, at

the surface, although $J(pNO_3^-)$ is strongest in summer, the addition of $NO_2$ is not maximum in summer due to low $pNO_3^-$ concentrations (Figure R1).

[Figure]

Figure R1. Simulated vertical profiles of nitrate aerosols (left), photolysis rate of nitrate aerosols (middle), and the $NO_2$ addition from nitrate photolysis (right) in January, April, July, and October 2017 over the contiguous US (CONUS).

**Specific comments:**

1. **L169-171: please indicate for which wavelengths the photolysis of particulate nitrate occurs.**
   Within the ultraviolet spectral region (>290 nm), the absorption spectrum of $pNO_3^-$ peaks around 302 nm (Gen et al., 2022). We have now specified this in the revised manuscript.

2. **L296: I propose to (also) cite the study by Castellanos et al. [2015] here, who showed that AMF calculations that explicitly account for aerosol absorption and scattering are on average 10% and up to a factor 2 higher than AMF calculations with implicit aerosol corrections.**
   We now cite this paper in section 5.

References:
Castellanos, P., Boersma, K. F., Torres, O., and de Haan, J. F.: OMI tropospheric $NO_2$ air mass factors over South America: effects of biomass burning aerosols, Atmos. Meas. Tech., 8, 3831–3849, https://doi.org/10.5194/amt-8-3831-2015, 2015.
Gen, M., Liang, Z., Zhang, R., Go Mabato, B. R., and Chan, C. K.: Particulate nitrate photolysis in the atmosphere, Environmental Science: Atmospheres, 2, 111-127, 10.1039/D1EA00087J, 2022.
Shah, V., Jacob, D. J., Dang, R., Lamsal, L. N., Strode, S. A., Steenrod, S. D., Boersma, K. F., Eastham, S. D., Fritz, T. M., Thompson, C., Peischl, J., Bourgeois, I., Pollack, I. B., Nault, B. A., Cohen, R. C., Campuzano-Jost, P., Jimenez, J. L., Andersen, S. T., Carpenter, L. J., Sherwen, T., and Evans, M. J.: Nitrogen oxides in the free troposphere: implications for tropospheric oxidants and the interpretation of satellite NO2 measurements, Atmos. Chem. Phys., 23, 1227-1257, 10.5194/acp-23-1227-2023, 2023.

**Response to comments by Referee #2**

In order to clarify the causes of the flattening of OMI $NO_2$ columns over the US after 2009, despite the decreased anthropogenic $NO_x$ fluxes, Ruijun Dang et al. investigate the sources and trends of background $NO_2$ levels over the US. In particular, the paper explores the impact of aerosol nitrate photolysis, aircraft emission trends, and wildfires. The major finding is that the inclusion of aerosol nitrate photolysis in the simulations leads to an annual increase of 13% of the model $NO_2$ column, and to a 7% decrease of the OMI $NO_2$ column when using the shape factor (AMF) re-calculated based on the model profiles. Both changes contribute to improve the model agreement with the data. The concomitant increasing trend of aircraft emissions and decline of surface anthropogenic NOx emissions leads to an increase of the AMF over the years (by about 10% between 2009 and 2017).

However, it is not clear from the paper whether the proposed effects do quantitatively explain the observed near-absence of $NO_2$ column trend seen in the OMI dataset. The paper builds upon recent papers by the same group: in particular, Shah et al. (2022) had already included particulate nitrate photolysis in the GEOS-Chem model. The innovative aspects of the present study work are (i) the use of the model shape factors (accounting for aerosol nitrate photolysis) to recalculate OMI $NO_2$ columns for 2017; (ii) the estimated effect of the increasing aircraft emissions on the AMFs. This study underscores the importance of the free tropospheric background when interpreting satellite $NO_2$ data. The article could be published if the following concerns are adequately addressed in a revised version.

We thank the reviewer for their thoughtful comments. Our response to the comments is as follows:

**Major comments:**

1. **A careful model evaluation against particulate nitrate observations is missing but is indispensable. We lack information on uncertainties regarding this important parameter. Does the model skill improve when accounting for the photolysis reaction? Could you provide maps of the distributions of aerosol nitrate from GEOS-Chem?**
   We have now added a discussion of the $pNO_3^-$ simulation evaluation in the revised manuscript.

   The photolysis of $pNO_3^-$ does not change the $pNO_3^-$ concentration much (~1%, see Figure S1) because deposition dominates the sink. We have specified this in the revised manuscript.

2. **The enhancement factor (EF) which multiplies the $HNO_3$ photolysis frequency is only crudely dependent on aerosol composition. Obviously, this factor is very uncertain, and furthermore, the enhancement is very likely wavelength-dependent. The shape of the absorption cross section is likely to change considerably in the condensed phase. This should be acknowledged and discussed.**
   We now state the uncertainty in EF in the revised manuscript.

3. **In Table 1, the background sources of lightning, soils and fires are higher in 2017 than in 2009. These sources are very variable though. Could you provide information**

**on the variability of these sources between 2009 and 2017? I wonder whether the results of Figure 4 would change if another year were chosen (e.g. 2016).**
We have now added a description of emission variability in Section 2.

For question about Figure 4, please see our response to comment #4.

4. **Please include the results for 2009 in Fig.4 (or in a Supplement). It is important to estimate the difference between the two years. Does the temporal correlation between model and observations improve when including nitrate photolysis?**
Thanks for pointing this out. We have now added Figure S2 and Figure 5 and a brief description about the change in nitrate photolysis effect between 2009 and 2017 in the revised manuscript. The inclusion of nitrate photolysis improves the temporal correlation between GEOS-Chem and OMI $NO_2$.

5. **The initial goal was to explain the OMI $NO_2$ flattening after 2009. It is not at all clear that you reached your goal. The model should be run for both years with the appropriate emissions, and the resulting $NO_2$ columns should be compared with the AMF-corrected OMI data. Furthermore, you need to convince the reader that comparing the years 2009 and 2017 is sufficient to conclude on the trend, given the interannual variability.**
Wang et al., (2021) have explained the flattening of OMI $NO_2$ by an increase in lightning emissions over the 2009-2019 period. Our work instead focused on the effects of nitrate photolysis, aircraft, and fires. We ran simulations for both 2009 and 2017 and compared them to the AMF-corrected OMI data. To improve the clarity of our goal, we have made edits in the revised manuscript.

Our lightning + soil emission changes between 2009 and 2017 (Table1) is consistent with the statistically significant increasing trend found by Wang et al., (2021) for the period 2009-2019, which implies that this 2009-2017 change could represent a longer-term trend that's relevant to the flattening. We have now added this discussion in the revised manuscript.

*References:*
Wang, Y., Ge, C., Castro Garcia, L., Jenerette, G. D., Oikawa, P. Y., and Wang, J.: Improved modelling of soil NOx emissions in a high temperature agricultural region: role of background emissions on NO2 trend over the US, Environ. Res. Lett., 16, 084061, 10.1088/1748-9326/ac16a3, 2021.

---

## Author Response (AR2)

**Response to comments by Referee #2**

The revised version is greatly improved and responds to most of my concerns. There is however still one major issue. Concerning the impact of pNO3- photolysis, there is something that does not add up. The authors claim that "The photolysis of pNO3- does not change the pNO3- concentration much (~1%, see Figure S1) because deposition dominates the sink." However, their Fig. R1 (in the response to Reviewer#1) shows that the pNO3- photolysis rate is of the order of 1E-5 s-1 in the lower troposphere (except in January). How could such a fast sink rate be negligible in comparison with deposition? Unless I'm wrong, the deposition lifetime should be at least one day. There is something going on that requires further investigation. A possible explanation could be that the pNO3- sink is nearly exactly compensated by enhanced HNO3 conversion to the particulate phase. In which case, the pNO3- photolysis would have a big impact on gas-phase HNO3 concentrations. Is this the case, and if so, can you show the impact in the paper and evaluate whether this is reasonable and in line with observations?

Thanks for pointing this out. The $pNO_3^-$ photolysis rate over CONUS is indeed comparable to its deposition loss rate, which we have now corrected in our revised manuscript.

The reason for the limited impact of $pNO_3^-$ photolysis on $pNO_3^-$ concentration is that the loss of $pNO_3^-$ is compensated by fast oxidation of $NO_x$ back to $HNO_3$ because $pNO_3^-$ photolysis releases both $NO_x$ and OH (from HONO photolysis). We have now included this discussion in our revised manuscript.

**Response to comments by Referee #3**

**The authors provided an analysis of the decreasing trends in tropospheric NO2 columns over the US. They found that the consideration of aerosol nitrate photolysis can lead to an increase in model NO2 by 13% and a decrease in the retrieved NO2 by 7%, and the combined effects can lead to a 45% reduction in the difference between modeled and retrieved changes in the year of 2009 and 2017. The topic is interesting, and the analysis is helpful for a better understanding of free tropospheric NO2. I recommend the paper for publication after consideration of the points below.**

We thank the reviewer for their thoughtful comments. Our response to the comments is as follows:

**Comments:**

1. **There are large interannual variabilities in tropospheric NO2 columns. The conclusion derived in this analysis represents the impacts of aerosol nitrate photolysis in the years 2009 and 2017. It cannot represent the impact on the trends in the period of 2009-2017.**
   Thanks for pointing this out. To clarify, we have now made some edits in Section 3.

2. **Have the authors considered the location and time consistency in simulations and OMI observations when calculating the mean NO2 columns over the CONUS? The OMI instrument crosses the equator at a local time of 13:45, and lots of data are filtered by the quality filters. To provide an accurate comparison, we should only consider modeled NO2 matching the temporal and spatial locations of OMI observations.**
   Yes, we have. We now clarify this in the revised manuscript.

3. **Lines 176: "Shah et al. (2023) found that incorporating aerosol nitrate photolysis in GEOS-Chem largely corrected the model's underestimation of NOx over the oceans during the ATom aircraft campaign".**
   **I checked Shah et al. (2023) and found "GEOS-Chem reproduces the shape of the PSS-inferred NO2 profiles throughout the troposphere for SEAC4RS and DC3 but overestimates NO2 concentrations by about a factor of 2" in the Abstract. I assume I perhaps have made some misunderstanding, but it seems that the discussions are not consistent.**
   Shah et al. (2023) did a model comparison against several aircraft campaigns, including ATom over the oceans and SEAC$^4$RS and DC3 over the US. For clarification, we now add a discussion of Shah et al. (2023) to the revised manuscript.

4. **The calculation of the enhancement factor is coarse. Considering the low SSA, it suggests an enhancement by a factor of 10 everywhere in the middle and upper troposphere. Can we apply the same factor to simulate free tropospheric NO2 globally?**
   In the upper troposphere, our calculated enhancement factor (EF) is about 10-20 over land at northern mid-latitudes, and up to 100 over the oceans in the southern hemisphere (see Figure 5 of Shah et al., (2023)). These values are within the range of EF that we currently know from field and laboratory studies (Ye et al., 2017; Romer et al., 2018; Andersen et al., 2023) and can be applied globally to test their effects. However, as we stated in the

conclusion section, further studies are needed in the future as these rates are still very uncertain. To clarify, we have made some edits in Section 2.

**References:**

Andersen, S. T., Carpenter, L. J., Reed, C., Lee, J. D., Chance, R., Sherwen, T., Vaughan, A. R., Stewart, J., Edwards, P. M., Bloss, W. J., Sommariva, R., Crilley, L. R., Nott, G. J., Neves, L., Read, K., Heard, D. E., Seakins, P. W., Whalley, L. K., Boustead, G. A., Fleming, L. T., Stone, D., and Fomba, K. W.: Extensive field evidence for the release of HONO from the photolysis of nitrate aerosols, Sci. Adv., 9, eadd6266, doi:10.1126/sciadv.add6266, 2023.

Romer, P. S., Wooldridge, P. J., Crounse, J. D., Kim, M. J., Wennberg, P. O., Dibb, J. E., Scheuer, E., Blake, D. R., Meinardi, S., Brosius, A. L., Thames, A. B., Miller, D. O., Brune, W. H., Hall, S. R., Ryerson, T. B., and Cohen, R. C.: Constraints on Aerosol Nitrate Photolysis as a Potential Source of HONO and NOx, Environ. Sci. Technol., 52, 13738-13746, 10.1021/acs.est.8b03861, 2018.

Shah, V., Jacob, D. J., Dang, R., Lamsal, L. N., Strode, S. A., Steenrod, S. D., Boersma, K. F., Eastham, S. D., Fritz, T. M., Thompson, C., Peischl, J., Bourgeois, I., Pollack, I. B., Nault, B. A., Cohen, R. C., Campuzano-Jost, P., Jimenez, J. L., Andersen, S. T., Carpenter, L. J., Sherwen, T., and Evans, M. J.: Nitrogen oxides in the free troposphere: implications for tropospheric oxidants and the interpretation of satellite NO2 measurements, Atmos. Chem. Phys., 23, 1227-1257, 10.5194/acp-23-1227-2023, 2023.

Ye, C., Zhang, N., Gao, H., and Zhou, X.: Photolysis of Particulate Nitrate as a Source of HONO and NOx, Environ. Sci. Technol., 51, 6849-6856, 10.1021/acs.est.7b00387, 2017